# Effects of Vitamin C on the Gonad Growth, Texture Traits, Collagen Content and Synthesis Related Gene Expression of Sea Urchin (*Mesocentrotus nudus*)

**DOI:** 10.3390/ani14172564

**Published:** 2024-09-03

**Authors:** Haijing Liu, Panke Gong, Dan Gou, Jiahao Cao, Weixiao Di, Jun Ding, Yaqing Chang, Rantao Zuo

**Affiliations:** Key Laboratory of Mariculture and Stock Enhancement in North China’s Sea (Ministry of Agriculture and Rural Affairs), Dalian Ocean University, Dalian 116023, China; lhaijing@hotmail.com (H.L.); 15072959865@163.com (P.G.); gd1951875973@163.com (D.G.); 19818925590@163.com (J.C.); dwx980914@gmail.com (W.D.); dingjun1119@dlou.edu.cn (J.D.); yqchang@dlou.edu.cn (Y.C.)

**Keywords:** vitamin C, *Mesocentrotus nudus*, gonad growth, gonad texture, collagen content, collagen synthesis

## Abstract

**Simple Summary:**

The worth of sea urchin gonads in the market is determined by their specific features connected to size and texture. Compared with wild counterparts, farmed sea urchins fed formulated feeds showed faster gonad growth but worse gonad texture traits, such as adhesiveness, cohesiveness, hardness, springiness, chewiness, and gumminess. Thus, it is pressing to enunciate the underlying mechanisms and find solutions to solve this problem. As the most abundant collagen, collagen type I is momentous for improving the hardness, springiness, and chewiness of muscle as verified. Vitamin C is vital for participating in the regulation of hydroxyproline biosynthesis, which is an essential component of collagen. However, the effects of vitamin C addition on the collagen synthesis and underlying mechanisms remain poorly understood in any sea urchin species. Therefore, this study aims to investigate the impact of vitamin C on the gonad growth, texture, collagen content, and the expression of genes involved in the collagen synthesis of sea urchin (*Mesocentrotus nudus*). These results could contribute to precisely quantify the vitamin C addition in the diets to produce gonads with high texture quality in sea urchin.

**Abstract:**

The market value of sea urchin gonads is determined by the specific characteristics associated with gonad size and texture. Formulated feeds can effectively promote the gonad growth of sea urchins but cannot assure essential gonad texture traits. The objective of this study was to investigate the impact of vitamin C (VC) on the gonad growth, texture, collagen content, and the expression of genes involved in the collagen synthesis of sea urchins (*Mesocentrotus nudus*). Graded amounts of VC (0, 3000 and 6000 mg/kg) were supplemented to make three formulated feeds. Fresh kelp (*Saccharina japonica*) was used as the control diet. Each diet was randomly distributed to three tanks of *M. nudus*. The results indicated that the gonadosomatic index (GSI) and texture traits of *M. nudus* fed C3000 were significantly greater than those fed C0 and C6000. Collagen type I (Col I) in the gonads of *M. nudus* fed C3000 showed significantly greater areas than those fed C0 and C6000. Consistently, the expression levels of collagen alpha-1 (*colp1α*) of *M. nudus* fed C3000 were significantly higher than those fed C0 and C6000. As for the transforming growth factor beta (*tgf-β*)/Smads pathway, the expression levels of collagen synthesis genes (*tgf-β* receptor 1 and 2, smad nuclear-interacting protein 1 (*snip1*) and prolyl 4-hydroxylase subunit beta (*p4hβ*)) in the C3000 group were significantly greater than those in the C0, C6000 and kelp groups. On the contrary, the expression levels of collagen degradation genes (lysyl oxidase-like 2 (*loxl2*) and matrix metalloproteinase 14 (*mmp14*)) in the C3000 group were significantly lower than those in the C0, C6000 and kelp groups. In conclusion, VC at an addition level of 3000 mg/kg significantly increased the gonad texture and collagen contents of *M. nudus*, which could be accomplished by increasing collagen synthesis and inhibiting collagen degradation through the *tgf-β*/Smads pathway. These results could contribute to better understanding the beneficial effects of VC addition on the gonad texture quality of *M. nudus*.

## 1. Introduction

Sea urchin gonad, as the sole edible part, is widely praised for its unique flavor in some areas of the world, including Europe, Asia, North America, Australia and Chile [1,2]. But the overfishing coupled with global climate change and habitat destruction are the main factors for the seriously declining of wild sea urchin resources [3,4]. To cater to the extensive seafood market, it is crucial to engage in the large-scale aquaculture of sea urchins [5]. *Mesocentrotus nudus* is now considered to be one of the most promising sea urchin species [6,7,8], especially after the breakthrough in its large-scale seed production.

The worth of sea urchin gonads in the market is determined by size, color, taste and texture. Compared with wild counterparts, farmed sea urchins fed formulated feeds showed faster gonad growth but worse gonad texture traits, such as adhesiveness, cohesiveness, hardness, springiness, chewiness, and gumminess [9]. Thus, it is pressing to enunciate the underlying mechanisms and find solutions to solve this problem.

Collagen is the most abundant protein in the extracellular matrix (ECM) [10], which includes collagen type I (Col I), III, IV, V, XI and XVIII [11]. As the most abundant collagen, Col I is momentous for improving the hardness, springiness, and chewiness of muscle as verified in grass carp (*Ctenopharyngodon idellus*) [12], Atlantic salmon (*Salmo salar*) [13], and European sea bass (*Dicentrarchux labrax*) [14,15,16]. Furthermore, accelerating the deposition of collagen in the swim bladder improved its quality reflected by brittleness, chewiness, hardness, gumminess, and resilience in chu’s croaker (*Nibea coibor*) [17]. Studies on sea cucumber (*Apostichopus japonicus*) showed that collagen was an important factor in the maintenance of body wall texture [18].

Vitamin C (VC) is an essential micronutrient for maintaining the normal growth of aquatic animals [19]. It is vital for participating into the regulation of hydroxyproline (Hyp) biosynthesis, which is an essential component of collagen [20,21,22,23,24]. Adding an appropriate amount of VC to the feeds can effectively improve the collagen content and texture (hardness, springiness, chewiness, moisture content) in the muscle of large yellow croaker (*Larimichthys crocea*) [10], *C. idellus* [25] and black sea bass (*Centropristis striata*) [26]. The supplementation of VC (100 mg/kg) promoted the collagen synthesis and wound repair of California brown shrimp (*Penaeus californiensis*) [27]. However, the effects of VC addition on the collagen synthesis and underlying mechanisms remain poorly understood in any sea urchin species. Transforming growth factor beta (*tgf-β*)/Smads is one of the signaling pathways regulating collagen synthesis [28]. The activation of the *tgf-β*/Smads pathway increased the expression of collagen synthesis genes, such as *col1a1* and *col1a2* [29,30], and inhibited the expression of collagen degradation genes matrix metalloproteinase (*mmp*) [31]. The activity of lysyl oxidase (*lox*) and *mmp-2* in muscle of *L. crocea* significantly decreased as dietary VC increased [10]. Thus, it was hypothesized that VC can boost collagen production by increasing the expression of genes responsible for collagen synthesis and decrease the degradation of collagen through the *tgf-β*/Smads signaling pathway in *M. nudus.*

Therefore, the objective of this study was to investigate the impact of VC on the gonad growth, texture, collagen content and synthesis of *M. nudus.* It was aimed to explore (i) the effects of VC on the gonad growth of sea urchins, and the effects of the collagen content; (ii) the correlation between collagen content and gonad texture (hardness, springiness, chewiness, etc.); (iii) the mechanism of VC regulating the synthesis of sea urchin collagen and assess its potential as a functional additive for enhancing the growth and texture quality of *M. nudus* in aquaculture systems.

## 2. Materials and Methods

### 2.1. Ethics Statement

Because *M. nudus* are invertebrates, the Ethics Committee of Dalian Ocean University did not require this research to be authorized.

### 2.2. Experimental Diets

According to the findings of previous studies, the optimal VC was estimated to be 0.2–0.48% based on the gonad development and muscle texture of several fish species [17,22,27,32]. Therefore, VC was added at a level of 0%, 0.3% and 0.6% in this study. Three isoproteic (19.28%) and isolipidic (4.32%) formulated feeds were designed by adding graded levels (0, 0.3%, 0.6%) of VC (L-ascorbic acid). Vitamin-free casein (Gansu Hualing Dairy Co., Ltd., Gannan Tibetan Autonomous Prefecture, China) and gelatin were the main sources of protein. Table 1 provided a list of feed ingredients and the proximate composition of the diets used in this experiment. All solid feed materials were thoroughly ground to pass through a 280-micrometer sieve. The solid components for each diet were initially mixed. Following this, oils were incorporated evenly with the solid mixture. Subsequently, water was thoroughly blended with the mixture above. Once pelleted, the feed pellets (0.2 cm × 0.5 cm) were arranged on a drying tray and dried at a temperature of 50 °C. After drying, they were cooled down, packaged, and subsequently stored at a temperature of −20 °C. The dietary VC content was analyzed by high-performance liquid chromatography to be 1.07 (0 mg/kg), 2907 (3000 mg/kg), 5868 (6000 mg/kg), respectively.

### 2.3. Feeding Experiment

An 80-day feeding experiment was conducted at the experimental base of Dalian Ocean University (Dalian, China) from November 2023 to January 2024. Juvenile *M. nudus* were purchased from a breeding farm (Dalian, China). Following a 14-day period of acclimation, juvenile *M. nudus* (initial weight: 3.50 ± 0.20 g; test diameter: 20 ± 0.5 mm) were randomly assigned to 12 separate tanks (15 cm × 30 cm × 45 cm). If there was a notable variance in the initial total body weight among the tanks, *M. nudus* individuals would be swapped until the significance disappeared. Subsequently, each feed was randomly allocated to three separate tanks with each tank containing 15 *M. nudus*. The *M. nudus* were fed twice daily at 9:30 AM and 6:30 PM until they showed clear signs of satiation. The water was fully replaced every three days. During the feeding test, the water temperature was consistently kept within the range of 11 to 20 °C, the pH level was kept at 8.1 ± 0.1, the salinity was regulated at 31 ± 1‰, and dissolved oxygen was maintained above 7.5 mg L^−1^ by aeration.

### 2.4. Sampling Collection

Prior to sampling, all the *M. nudus* were starved for 36 h. After that, 9 sea urchins were weighed and dissected one by one in each tank. The *M. nudus* was carefully dissected and cut along the perioral membrane using sterile dissection. The digestive tract and coelomic fluid were carefully removed. The gonads were carefully taken out by using sterile spoons to maintain their integrity. The sampling was handled gently to avoid the damage of gonad texture. The gonads of each *M. nudus* were weighed to calculate the gonadosomatic index (GSI) as follows: GSI (%) = final gonad wet mass/final sea urchin weight × 100. Subsequently, the gonads of each *M. nudus* were divided into four parts, pooled, and used for later analysis. The first part was placed in formaldehyde for subsequent histological analysis to determine the distribution of collagen content. The second part was stored at −80 °C for further analysis of moisture content. The third and fourth parts were rapidly placed in liquid nitrogen for storage to measure the collagen protein content and analyze the expression of genes involved in collagen protein synthesis.

### 2.5. Feed Composition Analysis

AOAC methods were used to analyze and determine the nutritional components of the feed. The moisture content was assessed by calculating the loss of samples at 105 °C. The Kjeldahl method was employed to determine the crude protein content. The Soxhlet extraction method using ether as the solvent was used to determine the crude lipid content.

### 2.6. Gonad Moisture Content Analysis

The details about the determination of gonad moisture contents can be referred to Takagi et al. [33]. First, 1 g of *M. nudus* wet gonads was precisely weighed and placed into a glass Petri dish that has been previously dried to constant weight. Then, they were dried at 80 °C until they reach a constant weight. The moisture content of *M. nudus* gonads was calculated as follows: Moisture content = (1 − gonad dry weight)/1 × 100.

### 2.7. Determination of Gonad Collagen

The collagen content in sea urchin gonads was assessed by using the Hyp standard curve method [34]. The concentration of Hyp was analyzed by using commercial kits (A030-2-1, Nanjing Jiancheng Bioengineering Institute, Nanjing, China).

### 2.8. Microscopic Analysis of Collagen in the Gonad 

Hematoxylin and eosin (HE), Van Gieson (VG) and Sirius Red staining were used for histological analysis of the gonad tissues of *M. nudus*. The gonads were fixed in 4% paraformaldehyde at 4 °C for 24 h. After that, the fixed samples were dehydrated with a series of ethanol gradients (75~100%), xylene transparent, and embedded in paraffin; then, we prepared 4 μm thick gonad tissue sections with a Leica RM2016 microtome. After, we put the slices into xylene I 20 min, xylene II 20 min, absolute ethanol I 5 min, absolute ethanol II 5 min, 75% alcohol 5 min, and finally washed with water. Finally, two consecutive sections were stained for HE, VG, and Sirius Red staining, respectively.

HE staining: Firstly, the sections were put into hematoxylin staining solution for 3–5 min, followed by rinsing in water, differentiation with the appropriate solution, washing with tap water, restoration of blue color using a blue solution, and final rinsing under running water. Afterward, the sections underwent dehydration using 85% and 95% gradient alcohol, each for 5 min, and stained into eosin staining solution for 5 min. Finally, the slices were put into absolute ethanol I for 5 min, absolute ethanol II for 5 min, absolute ethanol III for 5 min, dimethyl I for 5 min, xylene II for 5 min transparent and neutral gum mounting.

VG staining: First, the sections were placed in VG staining solution for 1 min, and rinsed with water. Finally, the slices were put into absolute ethanol I for 5 min, absolute ethanol II for 5 min, absolute ethanol III for 5 min, xylene I for 5 min, xylene II for 5 min transparent and neutral gum mounting.

Sirius Red staining: First, we placed the slices in Sirius Red Stain for 8 min. Then, the slices were sequentially put into three cups of absolute ethanol for rapid rinsing and dehydration; the last time was slightly longer for 30S-2 min, and n-butanol was transparent for 10 s. Finally, the slices were sequentially put into xylene I for 5 min, xylene II for 5 min transparent and neutral gum mounting.

VG-stained sections were observed for tissue morphology and structure under a bright-field fluorescence microscope (Leica DM-4000, Leica, Nussloch, Germany). Sirius Red stained sections were observed for collagen protein distribution in gonadal tissues under a polarizing microscope (Nikon Eclipse E100, Nikon, Tokyo, Japan) [35]. The color images of Sirius Red stained sections were processed by Image-Pro Plus 6.0 software into grayscale images for calculation of Col I areas.

### 2.9. Texture Analysis

The texture analysis of the gonads was conducted using a texture analyzer (TMS-Pro, FTC, Sterling, VA, USA). For details, refer to the operation method of Martinez et al. [36]. Each sample in the experiment was compressed twice to 50% of its original height using a cylindrical probe at a speed of 30 mm/min (20 mm in diameter). Hardness (N) refers to the maximum peak force during initial compression of the sample; adhesiveness (N.mm) is the energy needed to separate a cylindrical probe from the sample; springiness (mm) indicates the extent of sample recovery after initial compression; cohesiveness (ratio) denotes the internal adhesion within the sample; gumminess (N) represents the viscosity properties (hardness × cohesiveness) of a semi-solid sample; and chewiness (mJ) measures the energy required to chew a solid sample (springiness × gumminess).

### 2.10. Real-Time Quantitative PCR (RT-PCR)

The expression of *colp1α*, *colp2α*, *colp3α*, *tgfβr1*, *tgfβr2,* smad nuclear-interacting protein 1 (*snip1*), *p4h* subunit beta (*p4hβ*), lysyl oxidase-like 2 (*loxl2*) and *mmp14* in *M. nudus* gonads were detected by using specific primers (Table 2). The RNA extraction and RT-PCR were performed following the procedures of Ning et al. [37].

### 2.11. Statistical Analysis

SPSS 26.0 software was used for data analysis. Firstly, the normality of the data distribution was checked by using Shapiro–Wilk’s test, and the homogeneity of variance was subjected to Levene’s test. Subsequently, the significance of differences among dietary groups was analyzed by using one-way ANOVA. When a significant difference (*p* < 0.05) was detected in statistical analysis, multiple comparisons were performed by using the Tukey test to evaluate significant variations in mean values across different dietary groups.

## 3. Results

### 3.1. The Impact of VC Supplementation on the GSI of M. nudus

The GSI of the formulated feed groups was significantly greater than that of the kelp group (*p* < 0.05). The GSI of the C3000 group was significantly greater than that of the C0 and C6000 groups (*p* < 0.05) (Table 3).

### 3.2. The Impact of VC Supplementation on Gonad Structure Characteristics of M. nudus

In the male gonads, spermatocyte exist along the follicle wall, and several droplets can be observed in nutritive phagocytes (NPs). In the female gonads, the early vitellogenic oocytes attaches along the inner side of the follicle wall, forming an irregular spherical shape. The gonads of female and male sea urchins in all diets remain in the growing and early stages of gametogenesis. There is no significant difference in the number of NPs and the follicles’ structural characteristics between male and female sea urchin gonads in the same diet group. In the formulated groups, the NPs in both male and female gonads of *M. nudus* increased as VC addition increased from 0 to 3000 mg/kg, and then they decreased with the continuous increase in VC. The gonads in both male and female *M. nudus* showed more NPs of the C3000 group than those of the C0 and C6000 groups. There were obviously fewer NPs in both male and female *M. nudus* gonads of the kelp group compared to the formulated feed groups. The follicles in both male and female *M. nudus* of the formulated feed groups were significantly bigger and more compact than those of the kelp group. The follicles in both male and female *M. nudus* of the C3000 group were significantly bigger and more compact than those of the C0 and C6000 groups (Figure 1 and Figure 2).

### 3.3. The Impact of VC Supplementation on Collagen Content in the Gonads of M. nudus

Col I fibers show strong birefringence, displaying colors from bright orange–red to pale yellow (Figure 3). The Col I area proportion in the gonads of the formulated feed groups was significantly greater than that of the kelp group (*p* < 0.05). The Col I area proportion in the gonads of the C3000 group was significantly greater than that of the C0 and C6000 groups (*p* < 0.05) (Figure 4a).

The collagen content of gonads in the formulated feed groups was significantly greater than that in the kelp group (*p* < 0.05). The collagen content of gonads in the C3000 group was significantly greater than those in the C0 and C6000 groups (*p* < 0.05) (Figure 4b,c).

### 3.4. The Impact of VC Supplementation on the Gonad Texture Quality in the Gonads of M. nudus

The hardness, gumminess, chewiness, and springiness of the formulated feed groups were significantly greater than those of the kelp group (*p* < 0.05). The hardness, cohesiveness, springiness, adhesiveness, chewiness, and gumminess of the C3000 group were significantly greater than those of the C0 and C6000 groups (*p* < 0.05) (Figure 5).

### 3.5. The Impact of VC Supplementation on the Moisture Content in the Gonads of M. nudus

The moisture content of the gonads in the kelp group was notably lower than that in the formulated feed groups (*p* < 0.05). The moisture content of gonads in the C3000 group was significantly greater than that in the C0 and C6000 groups (*p* < 0.05) (Figure 6).

### 3.6. The Impact of VC Supplementation on the Expression of Collagen Related Genes in the Gonads of M. nudus

The *colp1α*, *colp2α*, *colp3α*, *tgfβr1*, *snip1*, *tgfβr2*, *mmp14*, *p4hβ* and *loxl2* genes expression of *M. nudus* gonads were significantly affected by the VC in the diets (*p* < 0.05). The expression of *colp1α*, *colp2α*, *colp3α*, *tgfβr1*, *snip1*, *tgfβr2* and *p4hβ* genes of the C3000 group were significantly greater than those of the kelp, C0 and C6000 groups (*p* < 0.05). On the contrary, the expression of *mmp14* and *loxl2* genes of gonads in the C3000 group were significantly lower than those in the kelp, C0 and C6000 groups (*p* < 0.05) (Figure 7).

## 4. Discussion

The gonad of sea urchins is both a sex organ responsible for gametogenesis and a nutrient storage organ. Also, it is the sole edible part of sea urchins. It is widely acknowledged that the texture of sea urchin gonads is heavily influenced by the stage of gametogenesis [2]. The highest GSI and texture quality was usually observed at late stage III and early stage IV when gametes did not reach the state of maturation. The gametogenesis of sea urchins is characterized by fluctuating quantity of NPs [38]. In the present study, male and female *M. nudus* fed C3000 exhibited the highest GSI and quantity of NPs in their gonads among dietary groups. Sea urchin gonads with more NPs tend to exhibit higher hardness [39]. Many studies had shown that an appropriate amount of dietary VC can benefit the GSI of aquatic animals [40] with the optimal VC estimated for broodstock Japanese eel (*Anguilla japonica*) (1686 mg/kg) [22] and gold fish (*Carassius auratus*) (200 mg/kg) [41]. The texture quality between male and female sea urchin gonads is less pronounced during the growing and early maturation stage [2]. As the gametogenic cells matured and entered the late IV stage, female sea urchin gonads usually showed better texture quality, such as reddish, yellowish and springiness than female counterparts [2,42]. In this study, juvenile sea urchins were chosen as the research subjects because they were more sensitive to the nutrients in the diets. Although the gonad weight increased to a different extent among different dietary groups, they were mainly staying at the growing and early stages of gametogenesis [42]. Meanwhile, there was no significant difference in the number of NPs and follicles structural characteristics between male and female sea urchin gonads in the same diet group. Thus, the potential effects of sex on gonad texture quality can be neglected. However, it is worthwhile to further investigate the effects of VC on the gonad yield and quality between adult male and female sea urchins in the following studies to produce high-quality sea urchin gonads accepted by the consumers.

In this study, the gonad texture traits (hardness, cohesiveness, springiness, adhesiveness, and gumminess) of *M. nudus* in the C3000 group was significantly greater than those in the kelp, C0 and C6000 groups. Aussanasuwannakul et al. [43] have noted that the collagen content within the muscle tissue of *O. mykiss* is essential for sustaining cohesiveness. Previous research has confirmed that adding VC to the diet can enhance the fillet elasticity and cohesion of *L. crocea* [10]. It was previously found that the hardness and chewiness of sea cucumbers (*Apostichopus japonicus*) were positively correlated with the collagen contents in their body wall [44]. However, excess VC can inhibit the synthesis of Hyp, a characteristic amino acid of collagen, in human skin fibroblast [45]. In accordance, the findings of this study showed that 6000 mg/kg VC reduced the expression of genes involved in collagen synthesis. Collagen includes Col I, Col III, Col IV, Col V, Col XI, and Col XVIII [11]. Col I is typically present in dense fibers capable of enduring considerable mechanical stress and tension. It has been demonstrated that the hardness of fish fillet was positively correlated with the amount and spatial distribution of Col I [13]. In the present study, the greatest contents and area proportion of gonad Col I were observed in the C3000 group. Consistently, the gonad hardness in the C3000 group was significantly greater than that in the kelp, C0 and C6000 groups.

In this study, the moisture contents in gonads of the C3000 group were significantly greater than that in the kelp, C0 and C6000 groups. This aligns with the findings of a previous study which showed that the addition of 4800 mg/kg VC had significant promoting effects on the fillet hardness, adhesion, elasticity, and cohesion of hybrid sturgeon (*Acipenser schrenckii × Acipenser baeri*) [32]. This indicated that an appropriate addition of VC can effectively improve the collagen content and water-holding capacity of gonads, enhancing the gonad hardness. This was in accordance with the findings of Wei et al. [10], who found that the addition of VC significantly boosted the fillet hardness of *L. crocea* by increasing its collagen content and water-holding capacity. Huang et al. [27] also found that the dietary VC increased the muscle chewiness and hardness of Pacific white shrimp (*Litopenaeus vanname*) by increasing its collagen content and water-holding capacity. Gonads are the sole edible parts of sea urchins, and their texture quality directly affects the commercial value [46]. In this study, sea urchin juveniles rather than adults were chosen as the experimental objectives because juveniles grow faster and are more sensitive to the nutrients in the diets [47]. Indeed, the results of this study showed that the size, texture and collagen content of the gonads were significantly affected by dietary VC, which further verifies the effectiveness of the present experimental design. Nonetheless, it should be admitted that the gonad texture quality of juvenile sea urchins is somewhat different from that of adult counterparts. Thus, it is important to further investigate the effects of VC on the gonad yield and quality of adult sea urchins in the following studies. This can be beneficial for producing sea urchins with high-quality gonads through utilizing a suitable feeding strategy, such as adding a certain amount of VC to the diets.

It is acknowledged that the *tgf-β*/Smads pathway participates in the regulation of collagen synthesis [28]. Col I was encoded by the genes *colp1α* and *colp2α*, and Col III was encoded by the gene *colp3α*. In the present study, supplementing with 3000 mg/kg VC markedly enhanced the expression of *tgfβr1*, *tgfβr2* and *snip1* as well as *colp1α, colp2α* and *colp3α*. This indicated that VC could promote the expression of genes involved in collagen synthesis by activating the *tgf-β*/Smads pathway. Furthermore, activation of the *tgf-β*/Smads pathway can inhibit the collagen degradation and collagen crosslinking [31,48]. *Mmp14* is a critical gene that mediates the degradation of collagen [49]. *Loxl2* is an enzyme that can catalyze the collagen crosslinking in the ECM [50]. In this study, the addition of 3000 mg/kg VC significantly reduced the expression of *Mmp14* and *loxl2*. This indicated that an appropriate addition of VC can reduce collagen degradation and crosslinking and therefore increase collagen deposition in the gonads of *M. nudus*. *p4hβ* plays a vital role in the synthesis of collagens [51]. In this study, it was found that the addition of 6000 mg/kg VC significantly decreased the expression of *p4hβ* in the gonads of *M. nudus*. Thus, it was inferred that excessive VC can inhibit collagen synthesis by decreasing the expression of *p4hβ* that is regulated by the *tgf-β*/Smads pathway.

## 5. Conclusions

In conclusion, the addition of 3000 mg/kg VC to the feeds promotes GSI by increasing the number of NPs and moisture content in the male and female gonads of *M. nudus* juveniles. VC at an addition level of 3000 mg/kg increased the collagen content in the gonads of *M. nudus* by increasing the expression of collagen synthesis-related genes (*colp1α*, *colp2α*, *colp3α*, *tgfβr1*, *snip1*, *tgfβr2* and *p4hβ*) and decreasing the expression of collagen degradation-related genes (*mmp14* and *loxl2*) through the *tgf*-*β*/Smads pathway. It has been demonstrated that the texture quality of sea urchin gonads was positively correlated with the collagen content. These results can provide a preliminary understanding of the dose-dependent effects of VC addition on the production of gonads with high texture quality in this and other sea urchin species. Further research is needed to investigate the effects of VC on the texture quality of gonads in adult sea urchins between different genders.

## Figures and Tables

**Figure 1 animals-14-02564-f001:**
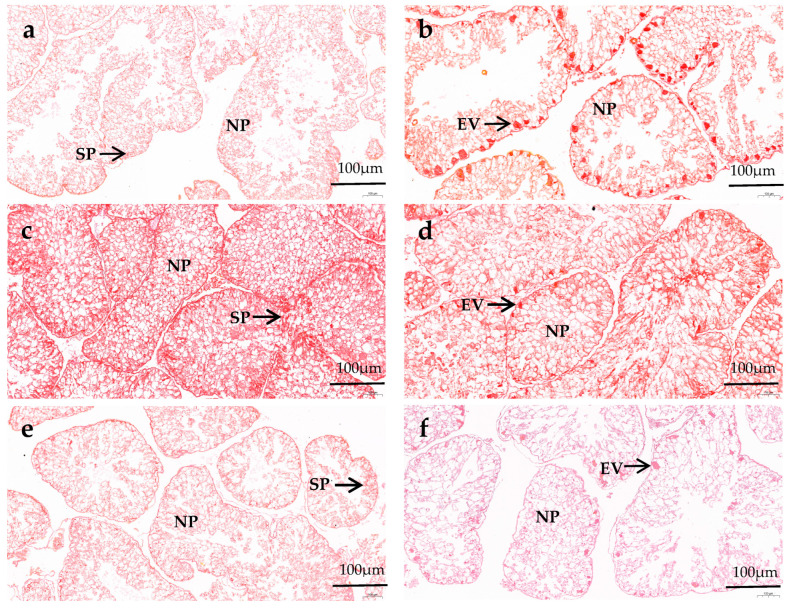
The gonad Van Gieson (VG) staining observations of juvenile sea urchin (*Mesocentrotus nudus*) fed diets with varying vitamin C (VC) concentrations (VC 0 mg/kg (C0), VC 3000 mg/kg (C3000), or VC 6000 mg/kg (C6000)). (**a**): C0, male; (**b**): C0, female; (**c**): C3000, male; (**d**): C3000, female; (**e**): C6000, male; (**f**): C6000, female; (**g**): kelp, male; (**h**): kelp, female. NP: nutritive phagocyte; SP: spermatocyte; EV: early vitellogenic oocytes.

**Figure 2 animals-14-02564-f002:**
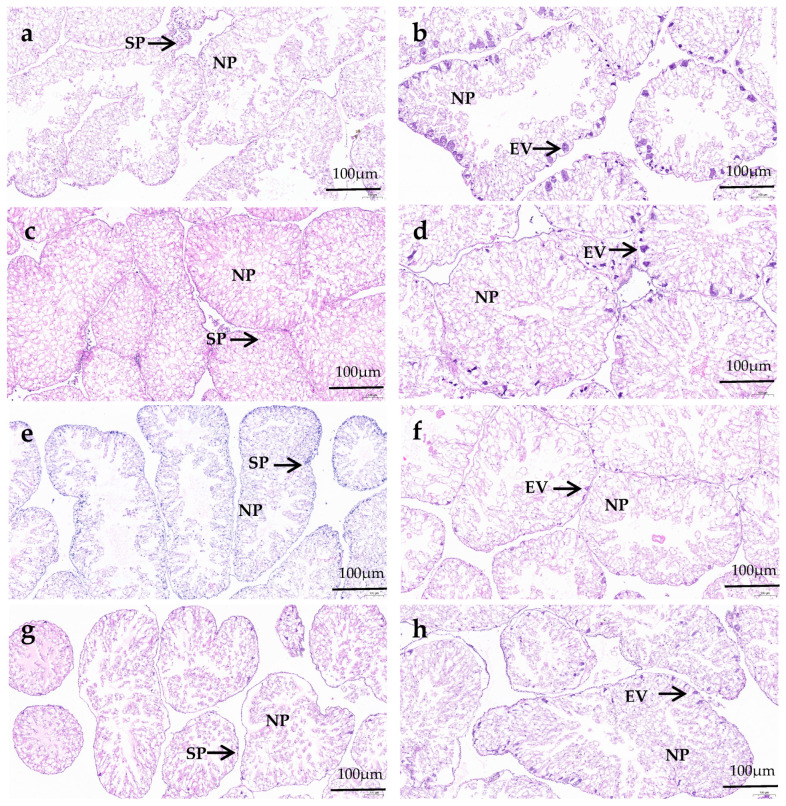
The hematoxylin and eosin (HE) staining observations of juvenile sea urchin (*Mesocentrotus nudus*) fed diets with varying vitamin C (VC) concentrations (VC 0 mg/kg (C0), VC 3000 mg/kg (C3000), or VC 6000 mg/kg (C6000)). (**a**): C0, male; (**b**): C0, female; (**c**): C3000, male; (**d**): C3000, female; (**e**): C6000, male; (**f**): C6000, female; (**g**): kelp, male; (**h**): kelp, female. NP: nutritive phagocyte; SP: spermatocyte; EV: early vitellogenic oocytes.

**Figure 3 animals-14-02564-f003:**
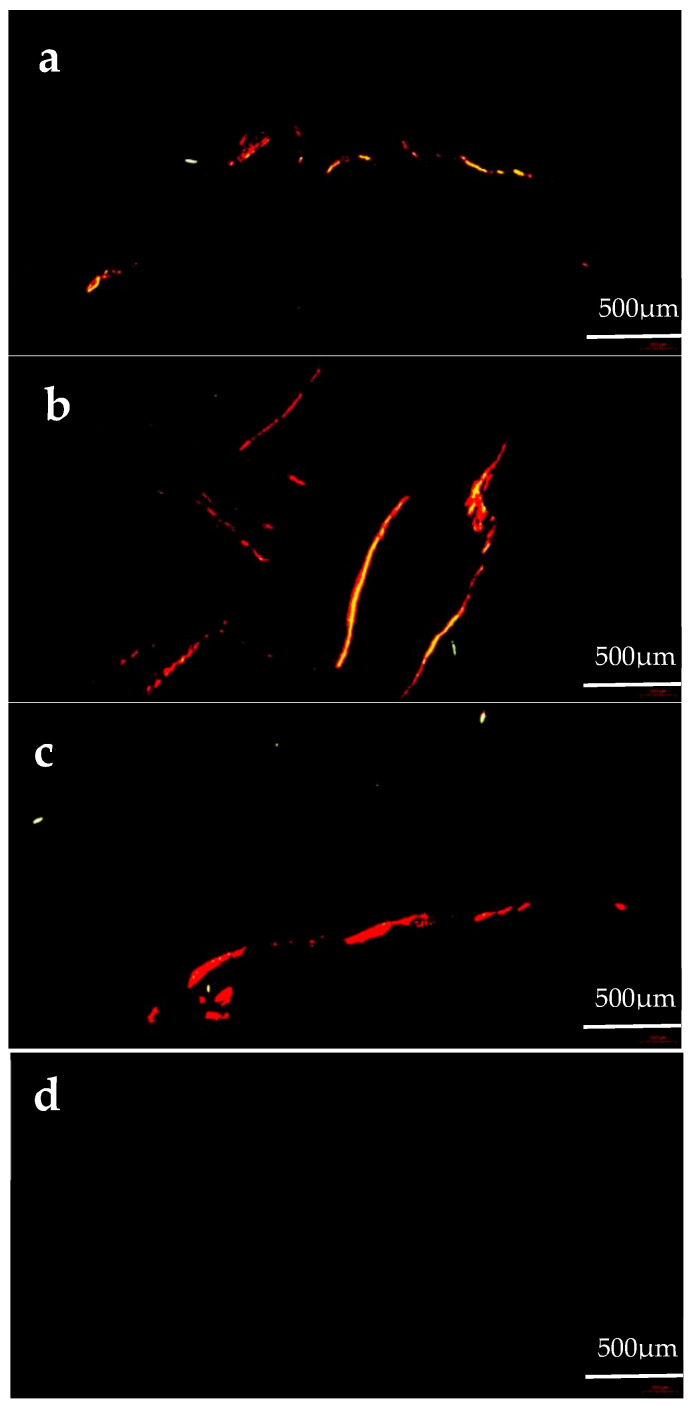
The gonad collagen content observations of juvenile sea urchin (*Mesocentrotus nudus*) fed diets with different vitamin C (VC) concentrations (VC 0 mg/kg (C0), VC 3000 mg/kg (C3000), or VC 6000 mg/kg (C6000)). (**a**): C0, Sirius Red stained; (**b**): C3000, Sirius Red stained; (**c**): C6000, Sirius Red stained; (**d**): kelp, Sirius Red stained.

**Figure 4 animals-14-02564-f004:**
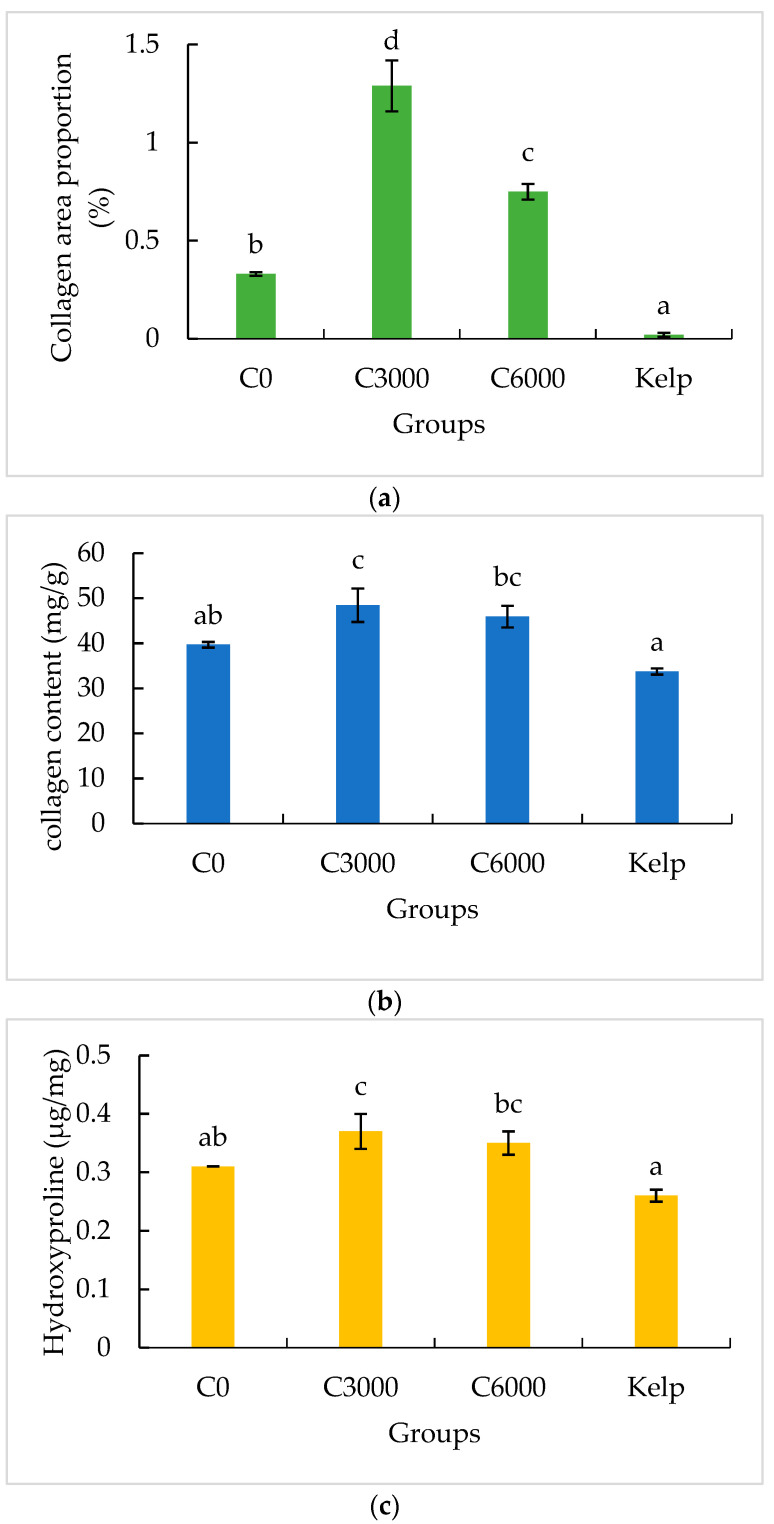
The gonad collagen area proportion (**a**), gonad collagen content (**b**) and gonad hydroxyproline (Hyp) content (**c**) of juvenile sea urchin (*Mesocentrotus nudus*) fed diets with varying vitamin C (VC) concentrations (VC 0 mg/kg (C0), VC 3000 mg/kg (C3000), or VC 6000 mg/kg (C6000)). Mean bars of the same cluster bearing with different letters indicate that they are significantly different at *p* < 0.05.

**Figure 5 animals-14-02564-f005:**
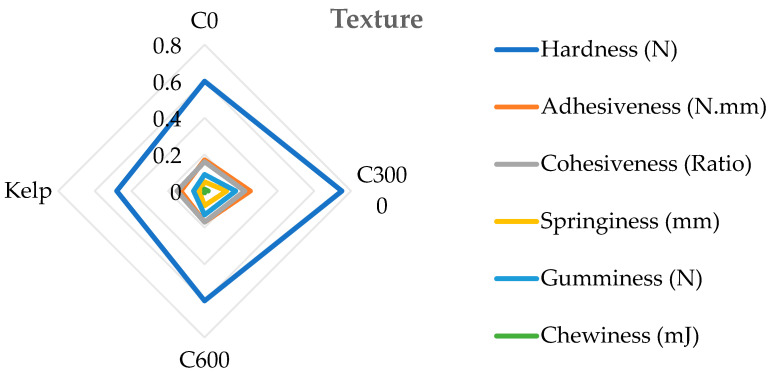
The gonad texture of juvenile sea urchin (*Strongylocentrotus nudus*) fed diets with varying vitamin C (VC) concentrations (VC 0 mg/kg (C0), VC 3000 mg/kg (C3000), or VC 6000 mg/kg (C6000)).

**Figure 6 animals-14-02564-f006:**
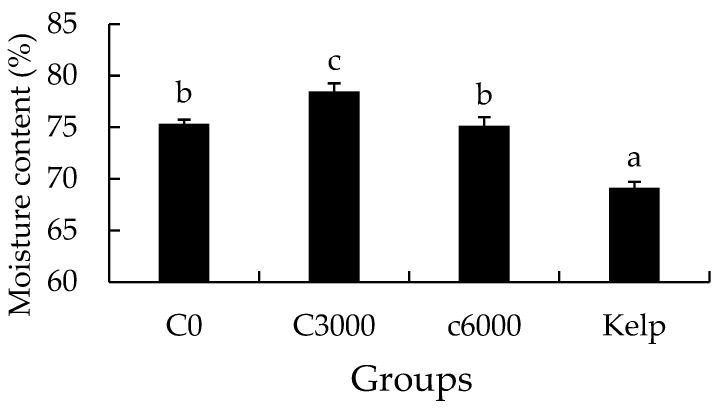
The gonad moisture content of juvenile sea urchin (*Mesocentrotus nudus*) fed diets with varying vitamin C (VC) concentrations (VC 0 mg/kg (C0), VC 3000 mg/kg (C3000), or VC 6000 mg/kg (C6000)). Mean bars bearing with different letters indicate that they are significantly different at *p* < 0.05.

**Figure 7 animals-14-02564-f007:**
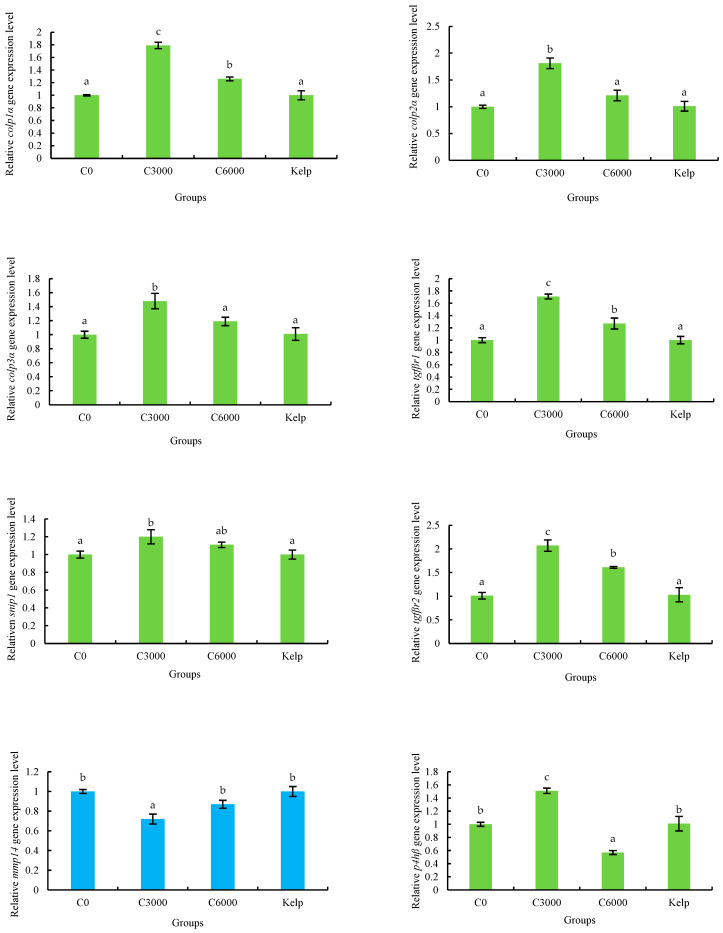
The gonad relative mRNA expression level of alpha-1 collagen (*colp1α*), *colp2α*, *colp3α*, transforming growth factor beta receptor type-1 (*tgfβr1*), *tgfβr2*, smad nuclear-interacting protein 1 (*snip1*), matrix metalloproteinase 14 (*mmp14*), prolyl 4-hydroxylase subunit beta (*p4hβ*) and lysyl oxidase-like 2 (*loxl2*) of juvenile sea urchin (*Mesocentrotus nudus*) fed diets with varying vitamin C (VC) concentrations (VC 0 mg/kg (C0), VC 3000 mg/kg (C3000), or VC 6000 mg/kg (C6000)). Mean bars bearing with different letters indicate that they are significantly different at *p* < 0.05.

**Table 1 animals-14-02564-t001:** Formulation and approximate composition of the experimental diets (% dry weight).

	Formulated Feeds with Varying Concentrations of Vitamin C		
Ingredients (%)	C0	C3000	C6000
Casein vitamins free ^1^	15.00	15.00	15.00
Gelatin ^2^	2.00	2.00	2.00
Wheat meal ^3^	20.00	20.00	20.00
Kelp meal ^4^	15.00	15.00	15.00
Microcrystalline cellulose	19.31	19.01	18.71
Braised corn flour ^5^	20.00	20.00	20.00
Vitamin C ^6^	0	0.3	0.6
Vitamin premix ^7^	2	2	2
Mineral premix ^8^	2	2	2
Calcium propionate	0.18	0.18	0.18
Choline chloride	0.1	0.1	0.1
Ethoxyquin	0.01	0.01	0.01
Soybean lecithin	1	1	1
Fish oil (FO)	2.4	2.4	2.4
Ca (H_2_PO_4_)_2_	1	1	1
Proximate composition			
Crude protein	19.26	19.25	19.23
Crude lipid	4.32	4.30	4.31
VC actual contents	0.0001	0.2907	0.5868

^1^ Casein vitamins free: crude protein 84.4% dry matter, crude lipid 0.6% dry matter; ^2^ Gelatin:crude protein 88.6% dry matter, crude lipid 0.5% dry matter; ^3^ Wheat meal: crude protein 13.88% dry matter, crude lipid 1.0% dry matter; ^4^ kelp meal: crude protein 19.31% dry matter; ^5^ Braised corn flour: crude protein 9.73% dry matter, crude lipid 4.12% dry matter; ^6^ Vitamin C: Ascorbic Acid as the vitamin C source (Nanjing Dulai Biotechnology Co., Ltd., Nanjing, China), the effective content is 99%; ^7^ Vitamin premix (mg or g kg^−1^ diet): vitamin D, 5 mg; vitamin K, 10 mg; vitamin B_12_, 10 mg; vitamin B_6_, 20 mg; folic acid, 20 mg; vitamin B_1_, 25 mg; vitamin A, 32 mg; vitamin B_2_, 45 mg; pantothenic acid, 60 mg; biotin, 60 mg; niacin acid, 200 mg; α-tocopherol, 240 mg; inositol, 800 mg; microcrystalline cellulose, 18.47 g; ^8^ Mineral premix (mg or g kg^−1^ diet): CuSO_4_·5H_2_O, 10 mg; Na_2_SeO_3_ (1%), 25 mg; ZnSO_4_·H_2_O, 50 mg; CoCl_2_·6H_2_O (1%), 50 mg; MnSO_4_·H_2_O, 60 mg; FeSO_4_·H_2_O, 80 mg; Ca (IO_3_)_2_, 180 mg; MgSO_4_·7H_2_O, 1200 mg; zeolite, 18.35 g.

**Table 2 animals-14-02564-t002:** Real-time *quantitative* PCR (RT-PCR) primers used in the present study.

Gene Abbreviation	Primer	Annealing Temperature (°C)	Amplicon Size (bp)	Amplification Efficiency (%)	Sequence Number
*colp1α* ^1^	F: TCAGTTCAGTGTCAGCGGATGTC R: ATGTTGCCTTCCAAGATGCCAATG	56	112	99	NM_214510.1
*colp2α* ^2^	F: GCACAGGTTCTTCTAAGCACAAGTC R: GTCATCACGCACGATACAAGCATAC	58	140	97	NM_214510.1
*colp3α* ^3^	F: CAGGCAGCAACAGGAAACGATAC R: ATGATGGTGGCGGTGATGATGG	59	141	98	NM_214466.1
*snip1* ^4^	F: GATAGGAGAGGCAATAGGCAGGAAC R: ACCTTCGTCTTCATCGTTTGTTTGG	58	138	105	XM_030995288.1
*tgfβr1* ^5^	F: AAGGTGATGAAGGAGTGCTGGTATC R: TGCGAGGCGTCACAGGTATTC	59	149	96	XM_793363.5
*tgfβr2* ^6^	F: GGTCATCGTCGTCTGTTCCGTAG R: ATGCTCGTGCTCTCCGTGTTG	56	150	97	XM_030972891.1
*mmp14* ^7^	F: CAGTGAGACTATGGCGATGATGAAC R: GGTCCTGTTGATGATCCTATAAGTGAG	59	145	102	NM_001033651.1
*p4hβ* ^8^	F: ATGGAGGAGGATGAGGAGATTGAC R: AGACTTGGGATGGACGCAGAC	59	141	101	NM_214532.1
*loxl2* ^9^	F: TCTTGTTGTCCTTCTTCCAGTTCTTC R: CAGTTCTCCTCAGCAGCACATTG	58	120	104	NM_001079547.1
*18s* ^10^	F: GTTCGAAGGCGATCAGATAC R: CTGTCAATCCTCACTGTGTC	58	145	96	D14365.1

^1^ Alpha-1 collagen (*colp1α*); ^2^ Alpha-2 collagen (*colp2α*); ^3^ Alpha-3 collagen (*colp3α*); ^4^ Smad nuclear-interacting protein 1 (*snip1*); ^5^ Transforming growth factor beta receptor type-1 (*tgfβr1*); ^6^ Transforming growth factor beta receptor type-2 (*tgfβr2*); ^7^ Matrix metalloproteinase 14 (*mmp14*); ^8^ Prolyl 4-hydroxylase subunit beta (*p4hβ*); ^9^ Lysyl oxidase-like 2 (*loxl2*); ^10^ 18S ribosomal RNA (*18s*).

**Table 3 animals-14-02564-t003:** Gonad growth performance in juvenile sea urchin (*Mesocentrotus nudus*) fed diets added with different concentrations of vitamin C (VC) ^1^.

	Feeds Added with Different Concentrations of Vitamin C			Kelp
	C0	C3000	C6000	
Gonad Weight (g)	1.30 ± 0.05 ^b^	1.47 ± 0.07 ^c^	1.16 ± 0.05 ^b^	0.35 ± 0.02 ^a^
Gonadosomatic Index (%)	17.45 ± 0.45 ^b^	19.50 ± 0.50 ^c^	16.33 ± 0.47 ^b^	3.88 ± 0.20 ^a^

^1^ Mean values in the same row with different superscript letters indicate significant differences at *p* < 0.05.

## Data Availability

The data of this study can be provided by the corresponding author upon request.

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
