# Peer review of "Effects of Vitamin C on the Gonad Growth, Texture Traits, Collagen Content and Synthesis Related Gene Expression of Sea Urchin (Mesocentrotus nudus)"

_animals, 2024, doi:10.3390/ani14172564_

Round 1

Reviewer 1 Report

Comments and Suggestions for Authors

This study clarified that the addition of 3000 mg/kg vitamin C to the feeds considerably promoted gonad size and texture quality, increased the collagen content by increasing the expression of collagen synthesis related genes, and decreasing the expression of collagen degradation in the gonads of Mesocentrotus nudus through well performed experimental design. It is therefore considered to be of high value to be published in “animals”.

However, it should be made clear that the sea urchins used in this study are juveniles, and it should be mentioned that the size, texture and collagen content of the gonads could be promoted in adult sea urchins to increase their commercial value in Discussion.

Specific comments are below.

Change “Strongylocentrotus nudus” to “Mesocentrotus nudus” in the text except for References.

Line 31. Correct “Nudus” to “nudus”.

Lines 66-67. It should be added that the worth of sea urchin gonads in the market is determined by not only size and texture but also color and taste.

Line 178. The background or basis why VC levels were determined as 0.3% and 0.6% should be defined.

Line 203. Test diameter of sea urchin should also be indicated. Sea urchins should be shown to be juveniles.

Lines 210-211. Was DO maintained above 7.5 mg L-1 by aeration?

Line 313. Statistical analyses of the normality and homoscedasticity of the data are necessary.

Line 323. Correct “Table 1” to “Table 3”.

Figure 1. Are the photographs of the gonad tissue at the same magnification? The scale bar should be indicated. In the photographs e, g and h, the readers cannot observe S (spermatozoa).

Line 335. Change “D” to “a-d”?

Line 336. Change “E-H” to “e-h”.

Figure 2. Are the photographs at the same magnification? The scale bar should be indicated.

Line 346. Change “A-D” to “a-d”.

Line 347. Change “E-H” to “e-h”.

Figure 3. The three figures should be the same size.

Figure 4. What units of 0, 0.2, 0.4, 0.6 and 0.8? This is not shown in Materials and methods.

Line 491. S. nudus should be changed to M. nudus juveniles.

Author Response

Response to reviewer #1:

  1. However, it should be made clear that the sea urchins used in this study are juveniles, and it should be mentioned that the size, texture and collagen content of the gonads could be promoted in adult sea urchins to increase their commercial value in Discussion.

We agree with the reviewer on this point. Although juvenile sea urchins can be used for a preliminary study to investigate the effects of VC on the gonad yield and quality, it would be better to further verify the results in adult sea urchins in the following studies. According to the reviewer’s suggestion, the discussion has been extended in the revised manuscript (Line 384-395) as follows:

Gonads are the sole edible parts of sea urchins, and their texture quality directly affects the commercial value [1]. In this study, sea urchin juveniles rather than adults were chosen as the experimental objectives because juveniles grow faster and are more sensitive to the nutrients in the diets [2]. Indeed, the results of this study showed that the size, texture and collagen content of the gonads were significantly affected by dietary VC, which further verify the effectiveness of the present experimental design. Nonetheless, it should be admitted that the gonad texture quality of juvenile sea urchins is somewhat different from that of adult counterparts. Thus, it deserves to further investigate the effects of VC on the gonad yield and quality of adult sea urchins in the following studies. This can be beneficial for producing sea urchins with high quality gonads through utilizing suitable feeding strategy, such as adding a certain amount of VC to the diets.

  1. Change “Strongylocentrotus nudus” to “Mesocentrotus nudus” in the text except for References.

According to the reviewer’s suggestion, “Strongylocentrotus nudus and S. nudus” has been changed to “Mesocentrotus nudus and M. nudus” in the revised manuscript.

  1. Line 31. Correct “Nudus” to “nudus”.

We are sorry for the mistake here. According to the reviewer’s suggestion, “Nudus” has been changed to “nudus” in the revised manuscript (Line 44).

  1. Lines 66-67. It should be added that the worth of sea urchin gonads in the market is determined by not only size and texture but also color and taste.

According to the reviewer’s suggestion, “The worth of sea urchin gonads in the market is determined by their specific features connected to size and texture.” has been changed to “The worth of sea urchin gonads in the market is determined by size, color, taste and texture.” in the revised manuscript (Lines 55-56).

  1. Line 178. The background or basis why VC levels were determined as 0.3% and 0.6% should be defined.

According to the reviewer’s suggestion, the background or basis why VC levels were determined as 0.3% and 0.6% has been defined in Materials & Methods of the revised manuscript (Line 99-101).

According to the findings of previous studies, the optimal VC was estimated to be 0.2% - 0.48% based on the gonad development and muscle texture of several fish species [3-6]. Therefore, VC was added at a level of 0%, 0.3% and 0.6% in this study.

  1. Line 203. Test diameter of sea urchin should also be indicated. Sea urchins should be shown to be juveniles.

According to the reviewer’s suggestion, “S. nudus (3.50 ± 0.20 g)” has been changed to “Following a 14-day period of acclimation, juvenile M. nudus (initial weight: 3.50 ± 0.20 g; test diameter: 20 ± 0.5 mm) were randomly assigned to 12 separate tanks (15 cm × 30 cm × 45 cm).” in the revised manuscript (Line 129-131).

  1. Lines 210-211. Was DO maintained above 7.5 mg L-1 by aeration?

Yes, dissolved oxygen was maintained above 7.5 mg L-1 by aeration. Thus, according to the reviewer’s suggestion, “and the dissolved oxygen concentration was maintained above 7.5 mg L-1” has been changed to “and dissolved oxygen was maintained above 7.5 mg L-1 by aeration” in the revised manuscript (Line 138-139).

  1. Line 313. Statistical analyses of the normality and homoscedasticity of the data are necessary.

According to the reviewer’s suggestion, "Firstly, the normality of the data distribution was checked by using Shapiro-Wilk test, and the homogeneity of variance was subjected to Levene’s test." has been added in the revised manuscript (Lines 230-232).

  1. Line 323. Correct “Table 1” to “Table 3”

We are sorry for the mistake here. According to the reviewer’s suggestion, “Table 1” has been changed to “Table 3” in the revised manuscript (Line 241).

  1. Figure 1. Are the photographs of the gonad tissue at the same magnification? The scale bar should be indicated. In the photographs e, g and h, the readers cannot observe S (spermatozoa).

Yes, photographs of these gonad tissues were taken at the same magnification. We are sorry for the confusion here. According to the reviewer’s suggestion, the scale bar has been added in all photographs of the revised manuscript.

Thanks to the reviewer’s for the reminder, in order to avoid misunderstandings during the reading process, the "S (spermatozoa)" marker has been removed from the revised manuscript.

  1. Line 335. Change “A-D” to “a-d”?

According to the reviewer’s suggestion, “A-D” has been changed to “a: C0, male; b: C0, female; c: C3000, male; d: C3000, female.” in the revised manuscript (Line 269 and 274).

  1. Line 336. Change “E-H” to “e-h”.

According to the reviewer’s suggestion, “E-H” has been changed to “e: C6000, male; f: C6000, female; g: kelp, male; h: kelp, female.” in the revised manuscript (Line 270 and 275).

  1. Figure 2. Are the photographs at the same magnification? The scale bar should be indicated.

Yes, photographs of these gonad tissues were taken at the same magnification. We are sorry for the confusion here. According to the reviewer’s suggestion, the scale bar has been added in all photographs of the revised manuscript.

  1. Line 346. Change “A-D” to “a-d”.

According to the reviewer’s suggestion, “A-D” has been changed to “ a: C0, Sirius Red stained; b: C0, Grayscale Image; c: C3000, Sirius Red stained; d: C3000, Grayscale Image.” in the revised manuscript (Line 288-289).

  1. Line 347. Change “E-H” to “e-h”.

According to the reviewer’s suggestion, “E-H” has been changed to “e: C6000, Sirius Red stained; f: C6000, Grayscale Image; g: kelp, Sirius Red stained; h: kelp, Grayscale Image.” in the revised manuscript (Line 289-290).

  1. Figure 3. The three figures should be the same size.

According to the reviewer's suggestion, the size of the three figures in the revised manuscript has been standardized to 59 × 96.01 mm.

  1. Figure 4. What units of 0, 0.2, 0.4, 0.6 and 0.8? This is not shown in Materials and methods.

Different texture indicators are represented by lines of different colors, and the units of the indicators are in parentheses in the Figure 4. Therefore, the units of 0, 0.2, 0.4, 0.6, and 0.8 are Hardness (N), Adhesion (N.mm), Springiness (mm), Cohesiveness (ratio), Gumminess (N), and Chewiness (mJ), respectively. These definitions have already been showed in the materials and methods (Line 209-216).

Figure 4. The gonad texture of juvenile sea urchin (Strongylocentrotus nudus) fed diets with varying vitamin C (VC)concentrations (VC 0mg/kg (C0), VC 3000mg/kg (C3000), or VC 6000mg/kg (C6000)).

  1. Line 491. nudus should be changed to M. nudus juveniles.

We are sorry for the mistake here. According to the reviewer’s suggestion, “S. nudus” has been changed to “In conclusion,the addition of 3000 mg/kg VC to the feeds promotes GSI by increasing the number of NPs and moisture content in the male and female gonads of M. nudus juveniles.” in the revised manuscript (Line 415-416).

References used in this response

  • Grosso, L.; Rakaj, A.; Fianchini, A.; Tancioni, L.; Vizzini, S.; Boudouresque, C.F.; Scardi, M. Trophic requirements of the sea urchin Paracentrotus lividus varies at different life stages: comprehension of species ecology and implications for effective feeding formulations. Frontiers in Marine Science. 20229, 865450. https://doi.org/10.3389/fmars.2022.865450.
  • Takagi, S.; Murata, Y.; Inomata, E.; Aoki, M.N.; Agatsuma, Y. Production of high quality gonads in the sea urchin Mesocentrotus nudus(A. Agassiz, 1864) from a barren by feeding on the kelp Saccharina japonicaat the late sporophyte stage. J Appl Phycol. 2019, 31, 4037–4048. https://doi.org/10.1007/s10811-019-01895-6.
  • Guo, H.; Liu, X.; Tian, M.; Liu, G.; Yuan, Y.; Ye, X.; Zhang, H.; Xiao, L.; Wang, S.; Hong, Y.; Sun, K. Effects of dietary collagen cofactors and hydroxyproline on the growth performance, textural properties and collagen deposition in swim bladder of Nibea coibor based on orthogonal array analysis. Rep. 2022, 27, 101375. https://doi.org/10.1016/j.aqrep.2022.101375.
  • Shahkar, E.; Yun, H.; Kim, D.J.; Kim, S.K.; Lee, B. I.; Bai, S. C. Effects of dietary vitamin C levels on tissue ascorbic acid concentration, hematology, non-specific immune response and gonad histology in broodstock Japanese eel, Anguilla japonica. 2015, 438, 115-121. https://doi.org/10.1016/j.aquaculture.2015.01.001.
  • Huang, T.; Guo, B.; Zheng, J.; Li, M.; Chen, Y.; Li, X.; Leng, X. Combined supplementation of hydroxyproline and vitamin C improved the growth and flesh quality of Pacific white shrimp (Litopenaeus vanname) cultured in low salinity water. Aquaculture and Fisheries. 2024.
  • Wu, Z.P.; Ruan, R.; Chen, H.; Chu, Z.P.; Li Y.; Yang W.J. Effects of dietary vitamin C supplemental level on growth performance, muscle quality and antioxidant indices of juvenile hybrid sturgeon (Acipenser schrenckii× Acipenser baeri). J-GLOBAL. 2020.

Reviewer 2 Report

Comments and Suggestions for Authors

The paper titled: “Effects of Vitamin C on the Gonad Growth, Texture Traits, Collagen Content and Synthesis Related Gene Expression of Sea Urchin (Strongylocentrotus nudus)” is interesting because it evaluates the effects of vitamin C (experimental diet) on a commercially relevant species. The experimental design is very simple, type: treatment/effect, without delving into the biological mechanisms. However, it helps to identify some data that may have an applicative impact.

I do not know if the journal allows it, but at the moment, the presence of the entire bibliography within the text, instead of simple (numerical) citations, makes the paper dispersive and difficult to read.

The authors should report in the introduction and throughout the paper any difference between male and female individuals. Also, from a commercial point of view, in various species of sea urchin there is a difference between the two sexes and this should also be taken into account in the materials and methods and in the results. Gonad texture quality is different for male or female.

For example authors report

“2.4. Sampling Collection

Prior to sampling, all the S. nudus were starved for 36 h. After that, 9 sea urchins were weighed and dissected one by one in each tank” .

Male or female gonads? In my opinion this information is crucial because different is the composition of organs due to presence of oocytes/eggs (cytoplasm, proteins and many mRNAs) or sperms (cells that have undergone removal of the cytoplasm).

Figure 1 should report a scale bar for each image.

Figure 1 reports in most cases male gonads, as indicated by the abbreviations: SP: spermatocyte; S: spermatozoa. The authors should better present these data by discussing the sense of presenting many photos of male gonads and the difference with female gonads.

The authors should report exactly the procedure used for the collection of the gonads. In sea urchins, mechanical stress induces the release of gametes into the external environment in the gonad. The emission of germ cells changes the content of the gonad itself.

The conclusions of this paper are too simple. The conclusions should be more augmented and should report references to male and female gonads.

Author Response

Response to reviewer #2:

  1. I do not know if the journal allows it, but at the moment, the presence of the entire bibliography within the text, instead of simple (numerical) citations, makes the paper dispersive and difficult to read.

We are sorry for the mistake here. According to the reviewer’s suggestion, the manuscript has been revised in strict accordance with the journal's formatting requirements.

  1. The authors should report in the introduction and throughout the paper any difference between male and female individuals. Also, from a commercial point of view, in various species of sea urchin there is a difference between the two sexes and this should also be taken into account in the materials and methods and in the results. Gonad texture quality is different for male or female.

We are grateful for the professional comments and suggestions on this point. Texture quality between male and female sea urchin gonads is less pronounced during the growing and early maturation stage [7]. As the gametogenic cells matured and entered late IV stage, female sea urchin gonads usually showed better texture quality, such as reddish, yellowish and springiness than female counterparts [7,8]. In this study, juvenile sea urchins were chosen as the research subjects because they were more sensitive to the nutrients in the diets. Although the gonad weight increased to different extent among different dietary groups, they were mainly staying at the growing and early stages of gametogenesis [8]. Meanwhile, there was no significant difference in the number of NPs and follicles structural characteristics between male and female sea urchin gonads in the same diet group. Thus, the potential effects of sex on gonad texture quality can be neglected. However, it deserves to further investigate the effects of VC on the gonad yield and quality between adult male and female sea urchins in the following studies to produce high quality sea urchin gonads accepted by the consumers.

Since the research objectives are juveniles, relevant description between male and female sea urchins cannot be added in the introduction and Materials & Methods. However, the discussion is extended from this perspective in the revised manuscript (Line 341-354) as follows:

Texture quality between male and female sea urchin gonads is less pronounced during the growing and early maturation stage [7]. As the gametogenic cells matured and entered late IV stage, female sea urchin gonads usually showed better texture quality, such as reddish, yellowish and springiness than female counterparts [7,8]. In this study, juvenile sea urchins were chosen as the research subjects because they were more sensitive to the nutrients in the diets. Although the gonad weight increased to different extent among different dietary groups, they were mainly staying at the growing and early stages of gametogenesis [8]. Meanwhile, there was no significant difference in the number of NPs and follicles structural characteristics between male and female sea urchin gonads in the same diet group. Thus, the potential effects of sex on gonad texture quality can be neglected. However, it deserves to further investigate the effects of VC on the gonad yield and quality between adult male and female sea urchins in the following studies to produce high quality sea urchin gonads accepted by the consumers.

  1. “2.4. Sampling Collection: prior to sampling, all the nudus were starved for 36 h. After that, 9 sea urchins were weighed and dissected one by one in each tank”. Male or female gonads? In my opinion this information is crucial because different is the composition of organs due to presence of oocytes/eggs (cytoplasm, proteins and many mRNAs) or sperms (cells that have undergone removal of the cytoplasm).

Yes, male and female sea urchin gonads differ in the composition of their organs due to the presence of oocytes/eggs or sperm. Thank you for pointing out the issue. While we agree that this is an important consideration, it is not appropriate to be included in this article. The juvenile sea urchins selected in this study were used as the research subjects, the weight of the sea urchins at the time of sampling was about 7 g. Sex is not easily distinguished at the time of sampling, and subsequent tissue sections can show that male and female gonad gametes have not yet differentiated. Texture quality between male and female sea urchin gonads is less pronounced during the growing and early maturation stage [7]. As the gametogenic cells matured and entered late IV stage, female sea urchin gonads usually showed better texture quality, such as reddish, yellowish and springiness than female counterparts [7,8]. In this study, juvenile sea urchins were chosen as the research subjects because they were more sensitive to the nutrients in the diets. Although the gonad weight increased to different extent among different dietary groups, they were mainly staying at the growing and early stages of gametogenesis [8]. Meanwhile, there was no significant difference in the number of NPs and follicles structural characteristics between male and female sea urchin gonads in the same diet group. Thus, the potential effects of sex on gonad texture quality can be neglected. Finally, thank you for your valuable suggestions. We will explore this suggestion further in a follow-up study of adult sea urchins.

  1. Figure 1 should report a scale bar for each image.

According to the reviewer’s suggestion, the scale bar has been added in all images of relevant figures.

  1. Figure 1 reports in most cases male gonads, as indicated by the abbreviations: SP: spermatocyte; S: spermatozoa. The authors should better present these data by discussing the sense of presenting many photos of male gonads and the difference with female gonads.

Thank you for pointing out the issue. Texture quality between male and female sea urchin gonads is less pronounced during the growing and early maturation stage [7]. As the gametogenic cells matured and entered late IV stage, female sea urchin gonads usually showed better texture quality, such as reddish, yellowish and springiness than female counterparts [7,8]. In this study, juvenile sea urchins were chosen as the research subjects because they were mainly staying at the growing and early stages of gametogenesis [8]. Meanwhile, the slice results of all male and female individuals were compared, it can be observed that the gametophytes are all at the same stage. Thus, the potential effects of sex on gonad texture quality can be neglected.

Thanks for the good suggestion of the reviewer. According to the reviewer’s suggestion, the relevant content has been reorganized in the revised manuscript as follows (251-276):

In the male gonads, spermatocyte exist along the follicle wall, and several droplets can be observed in nutritive phagocytes (NPs). In the female gonads, the early vitellogenic oocytes attaches along the inner side of the follicle wall, forming an irregular spherical shape. The gonads of female and male sea urchins in all diets remain in the growing and early stages of gametogenesis.There is no significant difference in the number of NPs and follicles structural characteristics between male and female sea urchin gonads in the same diet group. In the formulated groups, the NPs in both male and female gonads of M. nudus increased as VC addition increased from 0 to 3000 mg/kg, and then decreased with the continuous increase of VC. The gonads in both male and female M. nudus showed more NPs of C3000 group than those of C0 and C6000 groups. There were obviously fewer NPs in both male and female M. nudus gonads of kelp group compared to formulated feed groups. The follicles in both male and female M. nudus of the formulated feed groups were significantly bigger and more compact than those of kelp group. The follicles in both male and female M. nudus of C3000 group were significantly bigger and more compact than those of C0 and C6000 groups (Figure 1 and 2).

Figure 1. The gonad Van Gieson (VG) staining observations of juvenile sea urchin (Mesocentrotus nudus) fed diets with varying vitamin C (VC) concentrations (VC 0mg/kg (C0), VC 3000mg/kg (C3000), or VC 6000mg/kg (C6000)). a: C0, male; b: C0, female; c: C3000, male; d: C3000, female; e: C6000, male; f: C6000, female; g: kelp, male; h: kelp, female. NP: nutritive phagocyte; SP: spermatocyte; EV: early vitellogenic oocytes.

Figure 2. The Hematoxylin and eosin (HE) staining observations of juvenile sea urchin (Mesocentrotus nudus) fed diets with varying vitamin C (VC) concentrations (VC 0mg/kg (C0), VC 3000mg/kg (C3000), or VC 6000mg/kg (C6000)). a: C0, male; b: C0, female; c: C3000, male; d: C3000, female; e: C6000, male; f: C6000, female; g: kelp, male; h: kelp, female. NP: nutritive phagocyte; SP: spermatocyte; EV: early vitellogenic oocytes.

  1. The authors should report exactly the procedure used for the collection of the gonads. In sea urchins, mechanical stress induces the release of gametes into the external environment in the gonad. The emission of germ cells changes the content of the gonad itself.

In this study, juvenile sea urchins were chosen as the research subjects, they were mainly staying at the growing and early stages of gametogenesis in the gonads without the release of gametes [8]. The sampling was handled gently to avoid the damage of gonad texture without the release of gametes.

According to the reviewer’s suggestion, the process of collecting gonads has been added in the revised manuscript (Line 142-146), “Prior to sampling, all the M. nudus were starved for 36 h. After that, 9 sea urchins were weighed and dissected one by one in each tank. The sea urchin was carefully dissected and cut along the perioral membrane using sterile dissection. The digestive tract and coelomic fluid are carefully removed. The gonads were carefully taken out by using sterile spoons to maintain their integrity. The sampling was handled gently to avoid the damage of gonad texture. The gonads of each M. nudus were weighed to calculate the gonadosomatic index (GSI).”

  1. The conclusions of this paper are too simple. The conclusions should be more augmented and should report references to male and female gonads.

Thanks for the good suggestion of the reviewer. According to the reviewer’s suggestion, the relevant content has been reorganized in the revised manuscript as follows (Line 414-425):

In conclusion, the addition of 3000 mg/kg VC to the feeds promotes GSI by increasing the number of NPs and moisture content in the male and female gonads of M. nudus juveniles. VC at an addition level of 3000 mg/kg increased the collagen content in the gonads of M. nudus by increasing the expression of collagen synthesis related genes (COLP1α, COLP2α, COLP3α, TGFβR1, TGFβR2, SNIP1 and P4Hβ), and decreasing the expression of collagen degradation related genes (Mmp14 and LOXL2) through TGF-β/smads pathway. It has been demonstrated that the texture quality of sea urchin gonads was positively correlated with collagen content. These results can provide preliminary understanding of the dose-dependent effects of VC addition on the production of gonads with high texture quality in this and other sea urchin species. Further research is needed to investigate the effects of VC on the texture quality of gonads in adult sea urchins between different genders.

References used in this response

[7] Rocha, F.; Baião, L. F.; Moutinho, S.; Reis, B.; Oliveira, A.; Arenas, F.; Maia, M.R.; Fonseca, A.J.; Pintado, M.; Valente, L.M. The effect of sex, season and gametogenic cycle on gonad yield, biochemical composition and quality traits of Paracentrotus lividus along the North Atlantic coast of Portugal. Sci. Rep-Uk. 2019, 9 (1), 2994. https://doi.org/10.1038/s41598-019-39912-w.

[8] Phillips, K.; Hamid, N.; Silcock, P.; Delahunty, C.; Barker, M.; Bremer, P. Effect of season on the sensory quality of sea urchin (Evechinus chloroticus) roe. J. Food Sci. 2010, 75 (1), S20-S30. https://doi.org/10.1111/j.1750-3841.2009.01388.x.

Round 2

Reviewer 2 Report

Comments and Suggestions for Authors

The changes made by the authors and the clarifications reported can be considered satisfactory.

Author Response

Dear editor,

Thanks for your constructive comments. We have carefully considered and responded all the comments and made corresponding revisions. Please feel free to contact us if there is any thing for us to improve the quality of this manuscript.

Best regards

Rantao zuo

Response to Editor’s:

1、There are some errors in the submitted manuscript that need to be corrected: some links to bibliographic references are broken within the text and the following message appears in different parts of the ms "Error! Reference source not found". This needs to be corrected by authors.

We are sorry for the mistake here. According to the editor’s suggestion, the errors has been corrected in the revised manuscript.

2、Abbreviation of gene names should be in lowercase and italics. Please correct.

We are sorry for the mistake here. According to the editor’s suggestion, the abbreviation of gene names has been corrected inthe revised manuscript.

3、Table 2; include the efficiency of amplification, amplicon size and NCBI sequence number.

We are grateful for the professional comments and suggestions on this point. According to the editor’s suggestion, the Table 2has been extended in the revised manuscript (Line 223) as follows:

Table 2. Real-time Quantitative PCR (RT-PCR) primers used in the present study.

Gene abbreviation

Primer

Annealing temperature (℃)

Amplicon size (bp)

Amplification efficiency (%)

Sequence number

colp1α1

F: TCAGTTCAGTGTCAGCGGATGTC

R: ATGTTGCCTTCCAAGATGCCAATG

56

112

99

NM_214510.1 

colp2α2

F: GCACAGGTTCTTCTAAGCACAAGTC

R: GTCATCACGCACGATACAAGCATAC

58

140

97

NM_214510.1 

colp3α3

F: CAGGCAGCAACAGGAAACGATAC

R: ATGATGGTGGCGGTGATGATGG

59

141

98

NM_214466.1 

snip14

F: GATAGGAGAGGCAATAGGCAGGAAC

R: ACCTTCGTCTTCATCGTTTGTTTGG

58

138

105

XM_030995288.1

tgfβr15

F: AAGGTGATGAAGGAGTGCTGGTATC

R: TGCGAGGCGTCACAGGTATTC

59

149

96

XM_793363.5 

tgfβr26

F: GGTCATCGTCGTCTGTTCCGTAG

R: ATGCTCGTGCTCTCCGTGTTG

56

150

97

XM_030972891.1 

mmp147

F: CAGTGAGACTATGGCGATGATGAAC

R: GGTCCTGTTGATGATCCTATAAGTGAG

59

145

102

NM_001033651.1

p4hβ8

F: ATGGAGGAGGATGAGGAGATTGAC

R: AGACTTGGGATGGACGCAGAC

59

141

101

NM_214532.1 

loxl29

F: TCTTGTTGTCCTTCTTCCAGTTCTTC

R: CAGTTCTCCTCAGCAGCACATTG

58

120

104

NM_001079547.1

18s10

F: GTTCGAAGGCGATCAGATAC

R: CTGTCAATCCTCACTGTGTC

58

145

96

 D14365.1

1Alpha-1 collagen (colp1α); 2Alpha-2 collagen (colp2α); 3Alpha-3 collagen (colp3α); 4Smad nuclear-interacting protein 1 (snip1);5Transforming growth factor beta receptor receptor type-1 (tgfβr1); 6Transforming growth factor beta receptor receptor type-2 (tgfβr2);7Matrix metalloproteinase 14 (mmp14); 8Prolyl 4-hydroxylase subunit beta (p4hβ); 9Lysyl oxidase-like 2 (loxl2); 1018S ribosomal RNA (18s).

4. Figure 3: what additional information is provided by images in Gray Scale tones? not demonstrated, thus, I recommend to delete them and just leave the colour images in Fig. 3

Thank you for pointing out the issue. the "grayscale images" marker has been removed from the revised manuscript.

 In order to eliminate misunderstandings, some sentences have been added to the Materials and Methods sections (Line 203-207) in the revised manuscript as follows:

Sirius Red stained sections were observed for collagen protein distribution in gonadal tissues under a polarizing microscope (Nikon Eclipse E100, Nikon, Tokyo, Japan). The color images of Sirius Red stained sections were processed by Image-Pro Plus 6.0 software into grayscale images for calculation of Col I areas.

In order to eliminate misunderstandings, some sentences have been added to the Results sections (Line 280-284) in the revised manuscript as follows:

Col I fibers show strong birefringence, displaying colors from bright orange-red to pale yellow. The Col I area proportion in the gonads of the formulated feed groups was significantly greater than that of kelp group (P < 0.05). The Col I area proportion in the gonads of C3000 group was significantly greater than that of C0 and C6000 groups (P < 0.05) (Figure 3 and 4(a)).

5. Did authors measure the content of vitamin C in gonads from different experimental groups? this information needs to be provided

It is really a good suggestion for us to investigate the relationship between dietary VC and VC retention in the gonads. Since this study was primarily aimed to investigate the effects of dietary VC on the growth, gonad texture quality and collagen synthesis of Mesocentrotus nudus, the VC content in the gonads was not measured due to insufficient gonad samples of juveniles. According to the results of this study, it can be observed that the collagen content and collagen synthesis related gene expression were significantly affected by dietary VC. Thus, it can be inferred that VC contents in the gonads were significantly affected. We really appreciate you for the valuable suggestions. The following studies will be performed to clarify this point by using adult sea urchins.

6、Was vitamin C measured once the feeds were manufactured? the nominal inclusion levels does not necessarily mean that this is the real content of vitamin C in feeds since loses or changes may occur during feed manufacture and processing. This information needs to be included.

We are grateful for the professional comments and suggestions on this point. According to the editor’s suggestion, the real content of vitamin C in formulated feeds has been added in the revised manuscript (Line 113-115) as follows:

The dietary VC content were analyzed by high-performance liquid chromatography to be 1.07 (0 mg/kg), 2907(3000 mg/kg), 5868 (6000 mg/kg), respectively. 
